# Predictive Utility and Metabolomic Signatures of TG/HDL-C Ratio for Metabolic Syndrome Without Cardiovascular Disease and/or Diabetes in Qatari Adults

**DOI:** 10.3390/metabo15090574

**Published:** 2025-08-28

**Authors:** Noora Kano, Najeha Anwardeen, Khaled Naja, Asma A. Elashi, Ahmed Malki, Mohamed A. Elrayess

**Affiliations:** 1Military Medical City Hospital, Medical Services, Qatar Armed Forces, Doha P.O. Box 2352, Qatar; 2Biomedical Research Center, QU Health, Qatar University, Doha P.O. Box 2713, Qatar; n.anwardeen@qu.edu.qa (N.A.); khaled.naja@qu.edu.qa (K.N.); asma.elashi@qu.edu.qa (A.A.E.); 3Biomedical Science Department, College of Health Sciences, QU Health, Qatar University, Doha P.O. Box 2713, Qatar; ahmed.malki@que.edu.qa; 4College of Medicine, QU Health, Qatar University, Doha P.O. Box 2713, Qatar

**Keywords:** metabolic syndrome, TG/HDL-C, metabolomics, monoacylglycerols

## Abstract

**Background:** Metabolic syndrome (MetS) is a major risk factor for cardiovascular disease (CVD) and type 2 diabetes mellitus (T2DM), especially in Middle Eastern populations with a high metabolic burden. This study aimed to evaluate the predictive utility of different lipid ratios, including triglyceride-to-high-density lipoprotein cholesterol (TG/HDL-C), total cholesterol (TC)/HDL-C, low-density lipoprotein (LDL-C)/HDL-C, and non-HDL-C/HDL-C, for identifying MetS. In addition, we aimed to characterise the underlying metabolic dysregulation using the most predictive lipid ratio by comparing metabolomic profiles between high-risk (T3) and low-risk (T1) groups. **Method:** We conducted a cross-sectional study using data from 2179 Qatari adults without CVD and/or T2DM. The predictive value of each lipid ratio for MetS was compared. Untargeted metabolomics was performed to profile metabolic changes between T3 and T1. **Results:** After adjustment for age, sex, and BMI, TG/HDL-C showed the highest discriminative ability for MetS (AUC = 0.896, 95% CI: 0.88–0.91; OR = 4.36, 95% CI: 3.63–5.28, *p* < 0.0001). In pairwise AUC comparisons, TG/HDL-C outperformed LDL-C/HDL-C (*p* = 2.6 × 10^−4^, after correction for multiple comparisons), with no significant differences versus other ratios. The high-risk group exhibited raised levels of phosphatidylethanolamines, phosphatidylinositols, and diacylglycerols, and lower levels of sphingomyelins and plasmalogens. These lipid classes have been suggested to be implicated in insulin resistance and metabolic dysfunction. Elevated monoacylglycerols were identified in high-TG/HDL-C groups, representing a previously underreported pattern. **Conclusions:** The TG/HDL-C ratio showed a better association with MetS compared with other lipid ratios and was linked to distinct metabolomic signatures. These findings suggest potential value for early risk evaluation, but longitudinal and mechanistic studies are needed to confirm clinical applicability.

## 1. Introduction

MetS is a cluster of metabolic abnormalities, characterised by insulin resistance or glucose intolerance, abdominal obesity, dyslipidemia, and hypertension that collectively increase the risk of CVD, T2DM, and stroke [1,2,3,4]. Diagnostic criteria differ between organizations. The International Diabetes Federation (IDF) criteria are focused on central obesity, while the National Cholesterol Education Program Adult Treatment Panel III (NCEP ATP III) adopts a broader definition, allowing diagnosis even if abdominal obesity is not present [5,6].

MetS results from a complex interaction between genetic predisposition and lifestyle factors, including obesity and physical inactivity. It is further driven by chronic inflammation, endocrine dysfunction, and oxidative stress [7,8]. Early detection and intervention, primarily through lifestyle modification, are essential to slow disease progression and prevent cardiometabolic complications [9]. Globally, MetS is estimated at 25% of the adult population, with higher prevalence in the Middle East and North Africa (MENA) [10,11,12,13]. However, comprehensive regional data are limited [13]. This highlights the urgent need for simple, reliable, and accessible biomarkers to facilitate early detection and risk stratification.

Dyslipidemia, characterised by elevated TGs and reduced HDL-C, is a core component of MetS [14]. The TG/HDL-C ratio serves as a comprehensive indicator of dynamic changes in lipid metabolism, making it a robust predictor of MetS and cardiovascular risk. Evidence from many studies suggests that the TG/HDL-C ratio is a valuable clinical tool for predicting MetS, offering a simpler and cost-effective alternative to traditional multi-parameter diagnostic criteria for identifying individuals at risk [15,16,17]. However, TG/HDL-C and similar ratios may not fully capture the underlying metabolic disturbances of MetS, particularly across diverse populations.

Metabolomics enables the comprehensive profiling of small molecules (<1500 Da) in biological samples, capturing downstream metabolic responses to genetic, transcriptomic, proteomic, and environmental influences [18]. Given the complexity of MetS, metabolomics is an ideal tool for comprehensively characterising the underlying metabolic dysregulation and gaining deeper insight into its pathophysiology. Prior studies have identified alterations in amino acids, energy metabolites, and lipid classes such as triglycerides (TG), ceramides (Cer), phosphatidylcholines (PC), phosphatidylethanolamines (PE), and sphingomyelins (SM) in individuals with MetS [19,20,21,22,23]. These metabolomic signatures may provide valuable insights for risk stratification and early diagnosis of MetS.

However, most previous studies have independently evaluated lipid ratios and metabolomic profiles, limiting mechanistic insight and the detection of early metabolic alterations in high-risk groups. Since clinical lipid ratios are accessible biomarkers that may reflect underlying metabolic disturbances, integrating these with comprehensive metabolomic profiling offers a unique opportunity to elucidate the metabolic pathways and signatures driving MetS risk. To our knowledge, no study has integrated stratification by the most discriminative lipid ratio with untargeted metabolomic profiling to capture early metabolic changes underlying MetS, particularly in MENA populations where the burden of MetS is high.

This study aimed to investigate whether traditional lipid ratios, including TG/HDL-C, LDL-C/HDL-C, TC/HDL-C, and non-HDL-C/HDL-C, have predictive utility for identifying MetS in Qatari adults. To further explore underlying mechanisms, we compared T3 and T1 tertiles of the most discriminative ratio in an exploratory metabolomic analysis, maximising contrast to identify distinct metabolic profiles associated with early MetS. We hypothesized that this approach would reveal distinct metabolomic profiles associated with early MetS.

## 2. Materials and Methods

### 2.1. Data Source and Study Population

This study analysed data from the Qatar biobank (QBB), a prospective cohort study that collects extensive health information from Qatari nationals and long-term residents [24]. Data collection included a structured socio-demographic questionnaire, detailed medical history, and standardized clinical and laboratory assessments. All blood samples were collected after an overnight fast. The study protocol was approved by the Institutional Review Boards of the Qatar Biobank (QF-QBB-RES-ACC-00178). Written informed consent was obtained from each participant prior to enrollment in the study.

Data from 2998 individuals were retrieved from the Qatar Biobank (QBB), representing all participants with available metabolomics data at the time of analysis (Appendix A). To evaluate early metabolic alterations and diagnostic markers prior to overt disease, we excluded participants with diagnosed T2DM and/or CVD at baseline. The final cohort contained 2179 individuals from the general population, including MetS-positive and MetS-negative individuals without overt cardiometabolic disease.

### 2.2. Definition of Metabolic Syndrome

MetS was defined according to the criteria established by the IDF, in which central obesity is a prerequisite, plus any two or more of the remaining four risk factors. In this study, central obesity is defined by sex-specific waist circumference (WC) thresholds or a body mass index (BMI) ≥ 30 kg/m^2^, in which case central obesity is assumed; WC is not required, and hence, it was not collected. Additional risk factors include the following: (1) Elevated TG ≥ 150 mg/dL (1.7 mmol/L), or specific treatment for this lipid abnormality. Information on specific medications were not available for this dataset. (2) Reduced HDL-C: <40 mg/dL (1.03 mmol/L) in men or <50 mg/dL (1.29 mmol/L) in women. (3) Elevated blood pressure: systolic blood pressure (SBP) ≥ 130 mmHg or diastolic blood pressure (DBP) ≥ 85 mmHg. (4) Elevated fasting plasma glucose (FPG) > 5.6 mmol/L (100 mg/dL).

Since participants with known CVD and/or diagnosed T2DM were excluded from this study, elevated fasting blood glucose and high blood pressure were underrepresented among those classified as MetS-positive. Therefore, the definition of MetS in this cohort was driven primarily by central obesity in combination with elevated TG and/or reduced HDL-C.

### 2.3. Lipid Ratio Calculation

Four lipid ratios were calculated to assess their potential utility in predicting MetS including TG/HDL-C, TC/HDL-C, LDL-C/HDL-C and non-HDL-C/HDL-C. Lipid values were expressed in mmol/L, and the ratios were calculated directly from lipid measurements for each participant.

### 2.4. Biochemical Measurements

Serum TC, TG, and HDL-C were measured using automated enzymatic colorimetric assays (CHO-POD and GPO-POD methods) according to manufacturer protocols and standardized laboratory procedures [25,26,27]. LDL-C was calculated using the Friedewald equation for participants with TG ≤ 400 mg/dL; LDL-C values were not reported for individuals with TG > 400 mg/dL, as the Friedewald formula is not valid at higher concentrations and direct measurements were not available. All biochemical measurements were performed at the central laboratory of Hamad Medical Corporation, accredited by the College of American Pathologists.

Analytical performance of lipid assays targeted coefficients of variation and bias below 4%, consistent with international guidelines. Internal and external quality control measures were applied throughout the study.

Serum glucose was measured using an enzymatic colorimetric assay (glucose oxidase or hexokinase method) [28]. All biochemical assays were subject to rigorous quality control protocols, maintaining coefficients of variation below 5%.

### 2.5. Metabolomic Profiling

Untargeted metabolomic profiling of serum samples was performed according to established protocol by Metabolon as previously described in the literature [29,30]. Briefly, metabolite extraction and analysis were conducted using ultra-performance liquid chromatography (UPLC) (Waters ACQUITY UPLC system, Waters Corporation, Milford, MA, USA) coupled to high-resolution mass spectrometry (HRMS; Thermo Q-Exactive, Thermo Fisher Scientific, Inc., Waltham, MA, USA) equipped with a heated electrospray ionization source. Metabolites were identified by comparing the acquired spectra to a comprehensive reference library of authenticated chemical standards. Data quality was ensured through the use of stable isotope-labelled internal standards, the inclusion of quality control (QC) samples, and standardized pre-analytical procedures for sample collection, storage, and preparation.

### 2.6. Statistical Analysis

The distribution of clinical parameters was assessed using the Shapiro–Wilk test. Depending on normality, group comparisons were performed using either the Student’s t-test or Mann–Whitney U test, with results presented as mean (SD) or median (IQR) as appropriate. The predictive ability of each lipid ratio for MetS was evaluated using both unadjusted and adjusted logistic regression models (glm function in R), with MetS status as the binary outcome. Adjusted models included age, sex, and BMI as covariates. Predicted probabilities from these models were used to generate receiver operating characteristic (ROC) curves, and the area under the curve (AUC) was calculated to assess discriminative performance. Pairwise AUC comparisons were conducted using DeLong’s test. Statistical significance was defined as a two-tailed *p*-value < 0.05. To assess potential multicollinearity among the lipid ratios, a logistic regression model including all four ratios as predictors was fitted, and variance inflation factors (VIFs) were calculated.

We used the most predictive lipid ratio from the ROC/AUC analysis as a biomarker for MetS. Participants were divided into tertiles based on TG/HDL-C, namely T1 (lowest), T2 (intermediate), and T3 (highest). For metabolomic analyses, we compared T3 and T1 groups to identify the top metabolites differentiating MetS risk, excluding T2 to minimize potential overlap. Multivariate analyses were conducted in SIMCA (v.18), including principal component analysis (PCA) to assess data quality and orthogonal partial least squares-discriminant analysis (OPLS-DA) to examine metabolic profiles associated with high and low TG/HDL-C tertiles. Univariate linear regression analyses were performed in R (RStudio v.4.2.1) on log-transformed metabolite values, with metabolites as dependent variables (y) and TG/HDL-C tertile (low vs. high) as the independent variable (x), adjusting for age, sex, BMI, and principal components PC1 and PC2. *p*-values were adjusted for multiple testing using the False Discovery Rate (FDR) method. Finally, functional enrichment analysis was performed on metabolites meeting a stringent FDR threshold (<0.0001) using the Wilcoxon sum of ranks test.

## 3. Results

### 3.1. General Characteristics of the Study Population

A total of 2179 participants were included, of whom 368 (16.9%) were classified as MetS-positive according to the IDF criteria and 1811 (83.1%) were MetS-negative. Compared to MetS-negative individuals, those with MetS were older (median 42 vs. 33 years, *p* < 0.0001) and were predominately males (61% vs. 48%). The MetS group also had higher median BMI (31.8 vs. 27.0 kg/m^2^, *p* < 0.0001), waist circumference, and systolic/diastolic blood pressure (all *p* < 0.0001).

As expected, MetS-positive participants had higher TGs and lower HDL-C, resulting in significantly elevated TG/HDL-C, TC/HDL-C, LDL/HDL-C, and non-HDL/HDL-C ratios (all *p* < 0.0001). They also had higher fasting glucose and insulin levels (both *p* < 0.0001). Additional differences were observed in liver enzymes (ALT, AST) and ferritin concentrations (all *p* < 0.0001) (Table 1). Categorical data and complete participant characteristics are detailed in Appendix A.

### 3.2. ROC Curve Analysis

ROC analysis was conducted to evaluate the diagnostic performance of lipid ratios in predicting MetS. Among all ratios, TG/HDL exhibited the highest discriminatory power (AUC = 0.896, 95% CI: 0.88–0.91) in the adjusted model and demonstrated a strong association with MetS (adjusted OR = 4.356, 95% CI: 3.63–5.28, *p* < 0.0001) (Table 2). In unadjusted models, TG/HDL-C outperformed TC/HDL-C (AUC = 0.829, 95% CI: 0.81–0.85), LDL-C/HDL-C (AUC = 0.769, 95% CI: 0.75–0.79), and non-HDL-C/HDL-C (AUC = 0.829, 95% CI: 0.81–0.85) (Figure 1). Sensitivity analyses excluding BMI as a covariate produced similar results, with TG/HDL-C maintaining the highest discriminatory performance (Appendix A). Pairwise AUC comparisons using DeLong’s test (Table 3) confirmed the discriminative power of TG/HDL-C over other lipid ratios in unadjusted analyses (*p* < 0.0001 for all comparisons). After adjusting for age, sex, and BMI, TG/HDL-C remained significantly better after adjustment for multiple comparisons than LDL-C/HDL-C (Adjusted *p* = 2.64 × 10^−4^) but not significantly different from TC/HDL-C or non-HDL-C/HDL-C (Adjusted *p* = 0.497 for both).

In gender-stratified analyses adjusted for age and BMI, the TG/HDL-C ratio demonstrated the highest diagnostic performance for MetS among all lipid ratios in both males (AUC = 0.904, 95% CI: 0.885–0.923; OR = 3.00, 95% CI: 2.48–3.69; *p* < 0.0001) and females (AUC = 0.922, 95% CI: 0.897–0.947; OR = 3.43, 95% CI: 3.15–8.78; *p* < 0.0001) (Appendix A). For individual lipid markers, triglyceride was a strong predictor in males (AUC = 0.902, 95% CI: 0.882–0.921; OR = 4.27, 95% CI: 3.35–5.54; *p* < 0.0001), and both HDL-C (AUC = 0.896, 95% CI: 0.870–0.923; OR = 0.00138, 95% CI: 0.00041–0.00417; *p* < 0.0001) and triglyceride (AUC = 0.888, 95% CI: 0.857–0.918; OR = 12.05, 95% CI: 8.19–18.27; *p* < 0.0001) were strong predictors in females (Appendix A). However, the TG/HDL-C ratio remained the most robust overall marker in both genders.

### 3.3. Multivariate Metabolomics Analysis

Following the identification of TG/HDL-C as the most predictive lipid ratio for MetS, participants were stratified by TG/HDL-C tertiles. This approach enables the identification of metabolites that distinguish individuals at the extremes of MetS risk (low vs. high), while avoiding the dilution of group differences that may occur with binary or intermediate groupings. To explore metabolic differences by lipid phenotype, OPLS-DA was performed between the lowest and highest tertiles of TG/HDL. The model yielded one predictive and three orthogonal components with strong performance metrics (R^2^X = 20.8%, R^2^Y = 72.1%, Q^2^ = 67.4%), indicating distinct separation between tertiles (Figure 2a). The corresponding loading plot provided a visual representation of those metabolites with the most significant influence on differentiating high and low TG/HDL-C tertiles, allowing for identification of key variables associated with increased MetS risk (Figure 2b).

### 3.4. Univariate Metabolite Associations

Univariate linear regression identified several metabolites significantly associated with TG/HDL ratio tertiles after adjusting for age, sex, BMI, and principal components 1 and 2. The top 10 metabolites are presented in Table 4. Complete list of the significant metabolite associations shown in Appendix A.

### 3.5. Functional Enrichment Analysis

Pathway enrichment analysis was performed on all analysed metabolites to gain further insight into the biological mechanisms underlying the observed metabolite differences and provide a statistical summary of the most perturbed pathways. Pathway enrichment analysis based on ranked metabolites revealed significant perturbations in lipid metabolism pathways, particularly SM (*FDR* = 2.56 × 10^−9^), plasmalogens (*FDR* = 2.51 × 10^−6^), and PE species (*FDR* = 1.26 × 10^−4^) (Table 5).

## 4. Discussion

This is the first study to evaluate both the diagnostic performance and metabolomic signature of the TG/HDL-C ratio for identifying MetS among Qatari adults. Among the tested lipid ratios, TG/HDL-C demonstrated the strongest discriminatory performance for MetS. It significantly outperformed LDL-C/HDL-C in adjusted pairwise comparisons, while differences with TC/HDL-C and non-HDL/HDL-C were not statistically significant. Its predictive performance remained robust across both sexes in gender-stratified analyses. These findings suggest that TG/HDL-C may serve as a simple and accessible biomarker for early MetS risk stratification in this high-burden population.

Our findings are consistent with those reported in other populations, further supporting the diagnostic value of the TG/HDL-C ratio. For example, a cross-sectional study of 1276 elderly adults in China demonstrated that TG/HDL-C was a strong predictor of MetS even after adjusting for blood pressure, blood glucose, age, sex, and BMI (OR = 3.07, 95% CI: 2.402–3.924, *p* < 0.001) [31]. Similarly, the Birjand Longitudinal Aging Study (BLAS), which included 1356 elderly adults, reported that the TG/HDL-C ratio had the highest diagnostic accuracy (AUC = 0.92) among lipid ratios for predicting MetS [32]. Additionally, individuals in the highest TG/HDL-C quartile had markedly increased odds of MetS compared to the lowest quartile.

Beyond metabolic syndrome, TG/HDL-C has also demonstrated predictive value in other metabolic conditions. In a recent U.S.-based cross-sectional study using NHANES data, TG/HDL-C outperformed individual lipid markers such as TG and HDL-C in predicting metabolic dysfunction-associated steatotic liver disease (MASLD), achieving an AUC of 0.732 [33]. This association remained after adjustment for demographic, clinical, and biochemical variables. While MASLD was not the focus of our study, both conditions share common pathophysiological features such as insulin resistance, dyslipidaemia, and central obesity. The consistent performance of TG/HDL-C across these related metabolic conditions suggests its broader potential as a biomarker of systemic metabolic dysfunction. Additionally, a recent review highlighted the TG/HDL-C ratio as a robust predictor of adverse cardiovascular outcomes, including coronary artery disease, stroke, and peripheral artery disease, further reinforcing its clinical relevance as a simple, routinely available marker for cardiometabolic risk [34].

While TG/HDL-C demonstrated clinical effectiveness in predicting MetS, its integration with metabolomics in our study offered mechanistic insights into the underlying lipid dysregulation. Stratification by TG/HDL-C tertiles revealed distinct metabolic profiles, with higher ratios exhibiting elevated levels of PEs, PIs, MAGs, and DAGs, along with a reduction in sphingomyelins and plasmalogens. These findings deepen our understanding of metabolic syndrome aetiology and highlight a panel of lipid classes with potential as diagnostic or prognostic biomarkers. They also lay the groundwork for future studies evaluating the translational potential of targeting these specific lipid pathways in the prevention and management of MetS.

DAG and MAG are glycerolipids that serve as intermediates in the biosynthesis of other lipids. DAG functions as a bioactive signalling lipid that interferes with insulin signalling in both skeletal muscle and liver. The accumulation of DAG in these tissues is an established driver of lipid induced insulin resistance [35,36,37]. Mechanistically, DAG activates specific protein kinase C (PKC) isoforms, including PKCθ in muscle and PKCε in the liver, leading to impaired insulin receptor function, increased serine phosphorylation of IRS-1, and downstream attenuation of PI3K/Akt signalling. These disruptions result in reduced glucose uptake in muscle and impaired suppression of hepatic glucose production, central features of the metabolic syndrome phenotype. Lipidomic studies in MetS individuals reveal elevated DAG levels, especially DG (32:1), DG (34:1), DG (34:2), and DG (38:5), correlating positively with BMI and associated with obesity and insulin resistance [22,38]. While most studies focus on DAG, our findings suggest that MAG may also contribute to lipid dysregulation in MetS [22]. Given their role as intermediates in TG metabolism, elevated MAGs could represent early disturbances in lipid turnover and insulin sensitivity, potentially arising from increased lipolytic activity or impaired MAG metabolism.

PE and PI are major subclasses of glycerophospholipids. Both consist of a glycerol backbone attached to two fatty acid chains and a phosphate-containing head group. The head group for PE is ethanolamine and for PI is inositol. PEs are the second most abundant class of phospholipids in mammalian membranes and play a critical role in membrane fluidity and cellular signalling, while PI are key regulators of cell signalling pathways [39,40].

In a large population study, PEs were found to be elevated in insulin-resistant individuals, showing positive correlations with fasting glucose, insulin, C-peptide, and triglycerides, and an inverse association with HDL-C [41]. Furthermore, studies in non-obese individuals with insulin resistance and T2DM have demonstrated increased levels of glycerophospholipids, including PEs [42]. Interestingly, mechanistic studies in animal models showed that reduced PE can lead to the development of obesity, insulin resistance, and disruption of hepatic glucose metabolism and lipid homeostasis [43].

Experimental studies have also elucidated the role of PI in metabolic regulation. Inositol deficiency has been associated with hepatic lipid accumulation, suggesting a regulatory role for PI in hepatic lipid metabolism [44]. However, in a rat model of metabolic syndrome, dietary supplementation with PI reduced hepatic steatosis, improved insulin levels, and decreased liver inflammation [45]. Similarly, in a diet-induced obesity mouse model, oral PI administration led to reduced body weight gain and improved liver function [46]. In a human lipidomic study, specific PI species, including PI (32:1) and PI (40:6), were found to be positively correlated with insulin levels and HOMA-IR, indicating a potential association between specific PI species and insulin resistance [22]. A recent Mendelian randomization study showed that genetically higher PI levels were associated with a 17% increase in MetS risk, reinforcing the biological role of PI in metabolic disturbance [47].

SM is a type of sphingolipid enriched in the cell plasma membrane, which plays an important role in maintaining membrane integrity and facilitating cell signaling pathways. Numerous studies have suggested that elevated levels of sphingolipids, particularly ceramides and SM, are associated with increased risk of cardiovascular and metabolic diseases [48,49]. Hanamatsu et al. (2014) reported high levels of saturated SM species, specifically SM C18:0 and C24:0, in young obese adults (BMI ≥ 35). These were linked to markers of obesity, insulin resistance, liver issues, and lipid dysregulation [50]. However, findings regarding SM and metabolic syndrome vary across studies, largely due to differences in acyl chain composition, which influence the biological activity of SM. For instance, in a mouse model of obesity, SM C14 was markedly increased, whereas SM C22, C22:1, and C24 were reduced [51]. Similarly, a study examining the effects of acute exercise on serum SM demonstrated that specific SM species play distinct metabolic roles [52]. SM C18:0 was associated with insulin resistance and inflammation, while SM C14:0, C22:3, and C24:4 were related to insulin secretion and glucose tolerance. Collectively, these findings highlight the species-specific roles of SM in metabolism regulation.

Plasmalogens are a subclass of glycerophospholipids distinguished by a vinyl ether bond at the sn-1 position and an ester bond at the sn-2 position of the glycerol backbone [53]. This unique structural feature, particularly the vinyl ether linkage, contributes to their specialized biological roles in membrane architecture, antioxidant defence, and cellular signalling processes. Their susceptibility to oxidative cleavage by reactive oxygen and nitrogen species makes them key endogenous antioxidants [53,54]. Additionally, enzymatic degradation by phospholipase A2 and cytochrome c releases fatty acids that serve as precursors for lipid mediators. These degradation processes are triggered by oxidative stress and chronic inflammation, both hallmark features of metabolic dysfunction [55,56,57]. Consequently, oxidative stress can reduce plasmalogen levels, and plasmalogen depletion may further compromise cellular antioxidant defences while contributing to increased generation of the bioactive lipid mediators. Thus, reduced plasmalogen in MetS may both reflect and exacerbate the underlying oxidative stress and chronic inflammation. Our results align with previous studies where decreased plasmalogens were associated with increased cardiometabolic risk and the development of T2DM [58,59]. In addition, another study found a positive correlation between circulating plasmalogen with HDL-C and insulin sensitivity and negative correlation with TG, supporting their potential protective role in metabolic health [41].

While our results provide new insights into the pathophysiology of MetS and its metabolic signatures, it is important to recognize that the clinical consequences of metabolic syndrome extend beyond traditional cardiometabolic outcomes. MetS has been increasingly linked with arrhythmias and neuropsychiatric complications, including cognitive decline and depression, although the direction and strength of these associations may vary across populations and clinical contexts [60,61]. Our study offers several strengths, particularly the use of comprehensive metabolomic profiling in a large cohort. Importantly, we combined an accessible clinical marker, which is routinely available in clinical practice, with untargeted metabolomics to provide both practical and mechanistic insights into metabolic syndrome. However, several limitations should be considered when interpreting our findings. As a cross-sectional study, causal relationships cannot be established. Data on dietary habits and physical activity were not collected, and the MetS and non-MetS groups were not matched for demographic or clinical factors, so residual confounding may remain despite statistical adjustment. Although baseline laboratory assessments indicated that participants were apparently healthy, the presence of subclinical dyslipidemias cannot be entirely excluded. Data on lipid-lowering therapy were also not available, so potential influences on lipid indices and metabolomic profiles remain unaccounted for. Central obesity was defined using BMI ≥ 30 kg/m^2^ as a proxy, which may not fully capture adiposity. Although lean individuals with dyslipidemia may exist, the majority of our cohort was overweight or obese. Adjustment for BMI in the analyses may also introduce overadjustment bias, as it is part of the MetS definition. TG and HDL-C are part of MetS criteria, so their ratio overlaps with the outcome, potentially inflating discrimination statistics. The optimal AUC threshold for clinical application was not established, and future research should aim to identify and validate clinically relevant thresholds for effective risk stratification. Furthermore, the study sample was limited to Qatari adults, which may affect generalisability to other populations. Finally, the T1 vs. T3 comparison was employed primarily to maximise biological contrast in the exploratory metabolomics analyses, whereas the main conclusions are supported by and remain unchanged when TG/HDL-C was analysed as a continuous variable.

## 5. Conclusions

This study demonstrates that the TG/HDL-C ratio shows potential as a marker for predicting MetS among Qatari adults. By integrating metabolomic profiling, we identified that elevated TG/HDL-C is associated with specific alterations in lipid classes, including PE, PI, DAG, and plasmalogens, reflecting mechanisms consistent with insulin resistance and metabolic dysfunction. Notably, we also observed elevated MAG in the high-TG/HDL-C group, highlighting a finding that has been underreported in previous studies. These metabolomic findings offer new biological insights into the disturbances underlying elevated TG/HDL-C. However, they are exploratory and do not establish causality or immediate clinical utility. The TG/HDL-C ratio, as a simple and accessible biomarker, holds promise for improving early risk identification and potentially guiding preventive strategies for metabolic syndrome, particularly in high-risk populations. Nevertheless, further longitudinal studies in independent cohorts are necessary to confirm these results and to establish their clinical application. Future research should also explore whether combining metabolomic markers with lipid ratios could further enhance screening and early risk stratification for metabolic syndrome, ultimately supporting earlier detection and more targeted interventions.

## Figures and Tables

**Figure 1 metabolites-15-00574-f001:**
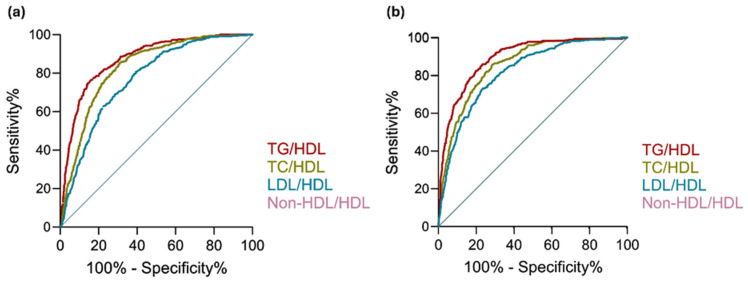
ROC curve analysis of lipid ratios for predicting MetS. (**a**) Unadjusted ROC curves for TG/HDL, TC/HDL, LDL/HDL, and non-HDL/HDL ratios. (**b**) ROC curves adjusted for age, gender, and BMI. TG/HDL showed the highest discriminatory performance in both models.

**Figure 2 metabolites-15-00574-f002:**
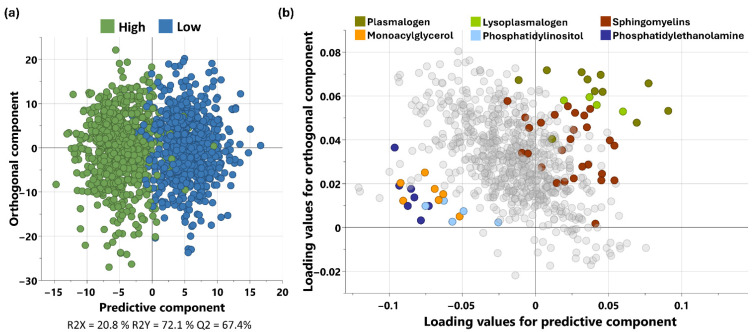
OPLS-DA analysis of low vs. high tertiles of TG/HDL ratio. The model provided 1 predictive component and 3 orthogonal components. (**a**) Scores plot from the OPLS-DA showing clear separation between the tertiles with R2X 20.8%, R2Y 72.1% and Q2 of 67.4%. (**b**) Loading plot showing all the metabolites with the enriched sub-pathways associated with the tertiles highlighted in different colours.

**Table 1 metabolites-15-00574-t001:** General characteristics of participants.

		MetS-Negative (*n* = 1811)	MetS-Positive (*n* = 368)	*p*-Value
**General characteristics**	Gender			
	Male	870 (48%)	225 (61%)	<0.001
	Female	941 (52%)	143 (39%)	
	Age	33 (27–43)	42 (34–50)	<0.001
	SBP (mmHg)	109 (101–118)	120.5 (111–133)	<0.001
	DBP (mmHg)	71 (65–77)	79 (71–87)	<0.001
	BMI (Kg/m^2^)	27 (23.85–30.48)	31.83 (28.66–35.23)	<0.001
	Weight (Kg)	74.3 (63.9–85.05)	88.8 (75.9–100.43)	<0.001
**Lipid profile**	TG/HDL-C	0.72 (0.47–1.08)	1.8 (1.32–2.62)	<0.001
	TC/HDL-C	3.46 (2.84–4.14)	4.83 (4.23–5.74)	<0.001
	LDL-C/HDL-C	2.1 (1.58–2.7)	3.04 (2.44–3.7)	<0.001
	NonHDL-C/HDL-C	2.46 (1.84–3.14)	3.83 (3.23–4.74)	<0.001
	NonHDL-C (mmol/L)	3.4 (2.84–4)	4.14 (3.55–4.73)	<0.001
	TG (mmol/L)	1 (0.73–1.34)	2 (1.4–2.58)	<0.001
	HDL-C (mmol/L)	1.39 (1.18–1.63)	1.06 (0.94–1.19)	<0.001
	LDL-C Calc (mmol/L)	3 (2.38–3.46)	3.17 (2.66–3.89)	<0.001
	TC (mmol/L)	4.8 (4.3–5.4)	5.2 (4.6–5.8)	<0.001
**Blood Sugar**	FBG (mmol/L)	4.9 (4.6–5.2)	5.6 (5–5.9)	<0.001
	Insulin (uU/mL)	8.1 (6–13)	17 (11.95–30)	<0.001
	C-Peptide (ng/mL)	2 (1.49–2.77)	3.33 (2.56–4.93)	<0.001
	HbA1C (%)	5.3 (5.1–5.5)	5.6 (5.3–5.8)	<0.001

All variables are presented as median (interquartile range). Comparisons between groups were performed using the Mann–Whitney U test. A *p*-value < 0.05 was considered statistically significant. Abbreviations: BMI, body mass index; HbA1C, glycated hemoglobin; HDL-C, high-density lipoprotein Cholesterol; LDL-C, low-density lipoprotein Cholesterol; FBG, Fasting Blood Glucose; SBP, Systolic Blood Pressure; DBP, Diastolic Blood Pressure. NonHDL-C, non-high-density lipoprotein cholesterol. TG, triglyceride. TC, total cholesterol.

**Table 2 metabolites-15-00574-t002:** Comparison of lipid ratios as predictors of metabolic syndrome using unadjusted and adjusted logistic regression models.

	Unadjusted Model					Adjusted Model				
	AUC	AUC 95% CI	OR	OR 95% CI	*p*	AUC	AUC 95% CI	OR	OR 95% CI	*p*
TG/HDL	0.873	0.85–0.89	4.134	3.53–4.88	<0.0001	0.896	0.88–0.91	4.356	3.63–5.28	<0.0001
TC/HDL	0.829	0.81–0.85	2.141	1.95–2.36	<0.0001	0.857	0.84–0.88	2.100	1.88–2.35	<0.0001
LDL/HDL	0.769	0.75–0.79	2.049	1.84–2.30	<0.0001	0.819	0.79–0.84	1.846	1.63–2.10	<0.0001
Non-HDL/HDL	0.829	0.81–0.85	2.141	1.95–2.36	<0.0001	0.857	0.84–0.88	2.100	1.88–2.35	<0.0001

Abbreviations: AUC, area under the receiver operating characteristic curve; OR, odds ratio; CI, confidence interval; TG, triglyceride; HDL, high-density lipoprotein cholesterol; TC, total cholesterol; LDL, low-density lipoprotein cholesterol; Non-HDL, non-high-density lipoprotein cholesterol; BMI, body mass index. Adjusted models include age, sex, and BMI as covariates.

**Table 3 metabolites-15-00574-t003:** Pairwise comparison of area under the curve (AUC) between TG/HDL and other lipid ratios for predicting metabolic syndrome.

	Unadjusted Model			Adjusted Model		
Comparison	AUC	*p*-Value	Adjusted *p*-Value	AUC	*p*-Value	Adjusted *p*-Value
TG/HDL	0.872	-	-	0.896	-	-
vs. TC/HDL	0.829	4.22 × 10^−7^	1.27 × 10^−6^	0.857	0.166	0.497
vs. LDL/HDL	0.769	2.2 × 10^−16^	6.6 × 10^−16^	0.819	8.8 × 10^−5^	2.64 × 10^−4^
vs. Non-HDL/HDL	0.829	4.22 × 10^−7^	1.27 × 10^−6^	0.857	0.166	0.497

Abbreviations: AUC, area under the receiver operating characteristic curve; TG, triglyceride; HDL, high-density lipoprotein; TC, total cholesterol; LDL, low-density lipoprotein; Non-HDL, non-high-density lipoprotein. Adjusted models include age, sex, and BMI as covariates.

**Table 4 metabolites-15-00574-t004:** Top 10 metabolites identified by univariate linear regression analysis as associated with low and high tertiles of TG/HDL ratio (adjusted for age, sex, BMI, and principal components 1 and 2).

Metabolite	Superpathway	Subpathway	Estimate	Std. Error	*p*-Value	FDR
**oleoyl-linoleoyl-glycerol (18:1/18:2) (2)**	Lipid	Diacylglycerol	0.739	0.033	1.51 × 10^−94^	7.57 × 10^−92^
**oleoyl-linoleoyl-glycerol (18:1/18:2) (1)**	Lipid	Diacylglycerol	0.766	0.034	1.70 × 10^−94^	7.57 × 10^−92^
**1-(1-enyl-palmitoyl)-2-oleoyl-GPC (P-16:0/18:1) ***	Lipid	Plasmalogen	−0.39	0.018	1.96 × 10^−87^	5.82 × 10^−85^
**1-stearoyl-2-linoleoyl-GPE (18:0/18:2) ***	Lipid	Phosphatidylethanolamine	0.547	0.028	1.47 × 10^−73^	3.26 × 10^−71^
**1-(1-enyl-palmitoyl)-2-linoleoyl-GPC (P-16:0/18:2) ***	Lipid	Plasmalogen	−0.331	0.017	8.27 × 10^−73^	1.47 × 10^−70^
**1-stearoyl-GPE (18:0)**	Lipid	Lysophospholipid	0.353	0.019	1.99 × 10^−70^	2.94 × 10^−68^
**1-palmitoyl-2-linoleoyl-GPE (16:0/18:2)**	Lipid	Phosphatidylethanolamine	0.549	0.03	2.76 × 10^−68^	3.50 × 10^−66^
**1-palmitoyl-2-oleoyl-GPE (16:0/18:1)**	Lipid	Phosphatidylethanolamine	0.576	0.032	4.92 × 10^−67^	5.46 × 10^−65^
**1-stearoyl-2-oleoyl-GPE (18:0/18:1)**	Lipid	Phosphatidylethanolamine	0.544	0.03	3.48 × 10^−66^	3.44 × 10^−64^
**1-linoleoylglycerol (18:2)**	Lipid	Monoacylglycerol	0.547	0.03	3.13 × 10^−65^	2.79 × 10^−63^

(1), (2) indicate distinct isomeric forms; (*) indicates a compound not confirmed by reference standard but confidently identified by Metabolon. Abbreviations: FDR, false discovery rate; GPC, glycerophosphocholine; GPE, glycerophosphoethanolamine.

**Table 5 metabolites-15-00574-t005:** Results from functional enrichment analysis based on metabolite ranks by *p*-value using the Wilcoxon sum of ranks test (FDR < 0.0001).

	*p*-Value	FDR
Sphingomyelins	2.81 × 10^−11^	2.56 × 10^−9^
Plasmalogen	5.51 × 10^−8^	2.51 × 10^−6^
Phosphatidylethanolamine	4.17 × 10^−6^	1.26 × 10^−4^
Monoacylglycerol	2.26 × 10^−5^	5.15 × 10^−4^
Phosphatidylinositol	2.41 × 10^−4^	4.39 × 10^−3^
Lysoplasmalogen	5.55 × 10^−4^	8.41 × 10^−3^

## Data Availability

The datasets used and/or analyzed during the current study are available from the corresponding author upon reasonable request.

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
