# Peer review of "Predictive Utility and Metabolomic Signatures of TG/HDL-C Ratio for Metabolic Syndrome Without Cardiovascular Disease and/or Diabetes in Qatari Adults"

_metabolites, 2025, doi:10.3390/metabo15090574_

Round 1
Reviewer 1 Report
Comments and Suggestions for Authors
This manuscript is an important contribution and I congratulate the authors for their work!
It leverages a large Qatari cohort, applies untargeted metabolomics, and demonstrates that the TG/HDL-C ratio outperforms other lipid ratios for identifying metabolic-syndrome cases. The data set is rich and the analyses are very robust, but several methodological clarifications, and a stronger discussion of confounding and circularity are essential before the results can be considered fully robust.
Revisions required
- Circularity between predictor and outcome
TG and HDL-C are themselves two of the five IDF criteria used to define MetS. Using their ratio to “predict” MetS therefore inflates discrimination statistics by design. Please acknowledge this limitation explicitly in the Discussion - Substituting BMI ≥ 30 kg m⁻² for waist circumference
The IDF definition requires central (abdominal) obesity, whereas BMI captures general adiposity and can misclassify muscular or short individuals. Add two-to-three lines in the Methods explaining why waist data were unavailable - Internal validation and confidence intervals for AUCs
All ROC analyses should be accompanied by 95 % confidence intervals. - NT-proBNP is lower in MetS but not discussed
The inverse association between NT-proBNP and MetS is counter-intuitive (many cardiometabolic studies find the opposite). Please devote a short paragraph to this result. - TyG index appears in Table 1 but is never analysed. Readers will expect at least a supplementary comparison, given TyG’s popularity as a MetS marker.
- Temper that elevated MAGs “represent a novel metabolic signature not reported by previous literature”. claim
This is an important claim. To support it, the authors could explicitly state that, to their knowledge, MAG elevation has not been described in MetS metabolomics. If possible, citing one or two key metabolomics that did not highlight MAGs would reinforce novelty. If no published metabolomics study has reported MAGs, a careful wording (e.g. “to our knowledge, this has not been previously reported”) is advised. Novelty claim appears in Abstract and Discussion . - Broader clinical context using new literature
MetS has systemic consequences that extend beyond traditional cardiometabolic endpoints, including arrhythmogenic and neuropsychiatric sequelae. You can add data from this article 10.3390/biomedicines11072012 - Abstract (lines 124-126): change “novel metabolic signature” to “previously under-reported monoacylglycerol pattern” unless you provide clear evidence of novelty.
- Table 4: it is dense; consider moving the full list to Supplementary material and showing only the top 10 metabolites in the main text. Clarify why some lipids appear twice.
- Confidence intervals for AUCs if possible.
Reviewer 2 Report
Comments and Suggestions for Authors
The manuscript by Kano et al. investigated the diagnostic performance of the TG/HDL‑C ratio for metabolic syndrome and examined associated metabolomic signatures in Qatari adults. This study provides potentially valuable insights into the topic. While the manuscript is informative and well‑structured, there are areas that would benefit from substantial revision to improve clarity, enhance methodological transparency, and ensure that the results, discussion, and conclusions are fully supported by the data. Detailed comments organized by key sections of the manuscript are provided below.
1. Title
The current title suggests that the study has demonstrated actual clinical utility of the TG/HDL‑C ratio. However, this is a cross‑sectional analysis that primarily evaluated diagnostic associations, not clinical application or intervention. A more cautious and precise title would better reflect the study’s scope. For example, consider phrasing such as “Diagnostic Performance of TG/HDL‑C Ratio…” or “Association of TG/HDL‑C Ratio with…”.
2. Abstract
a) As noted above, the study is cross‑sectional and evaluated associations and diagnostic performance primarily through ROC and AUC analyses. Therefore, statements in the Abstract referring to “clinical utility” or claims such as “facilitating earlier identification, more accurate risk stratification, and improved prognosis” are not fully supported by the data. Please temper the language to reflect diagnostic performance rather than clinical utility.
b) It would strengthen the Abstract to specify whether analyses were adjusted for confounders such as age, sex, and BMI.
c) Including key p‑values (for example, from ROC comparisons or regression results) would provide readers with a clearer sense of the statistical significance and strength of the associations.
3. Introduction
a) The Introduction devoted considerable space to general pathophysiology of MetS rather than focusing on the specific biomarker gap. Much of this background could be condensed. While the authors mentioned limited data in MENA populations and under‑explored integration of lipid ratios with metabolomics, the discussion of existing gaps is brief and inadequate. It seems that there is no clear summary of prior work integrating lipid ratios with metabolomics or a critical analysis of their limitations.
b) The justification for the study is somewhat generic. Although an “urgent need for simple biomarkers” is noted, the authors do not clearly explain why existing evidence on TG/HDL‑C is insufficient or how metabolomics would add value.
c) The metabolomics background lacks references to key findings, such as lipid classes previously linked to MetS and known limitations of prior studies. This makes the rationale less compelling. The authors should explain why metabolomics is needed after establishing TG/HDL‑C as a predictor and what actionable insights are expected.
d) Consider revising the final paragraph of the Introduction to clearly state the study objectives or hypotheses in a concise and organized manner.
4. Materials and Methods
Overall, this section is well‑structured and appropriate for the research questions. While this reviewer is not a statistician, the statistical methods appear generally suitable, and it is understood that the authors are responsible for the accuracy of the data analysis. Nevertheless, several aspects related to transparency and methodological rigor would benefit from further clarification:
a) For the logistic regression analyses, did the authors assess multicollinearity among the lipid ratios? Was model goodness‑of‑fit evaluated?
b) Logistic regression analyses were unadjusted, whereas linear regression models were adjusted for confounders (lines 149-151). Please clarify the rationale for this difference.
c) The authors stratified TG/HDL‑C into tertiles (T1, T2, T3) and compared T1 with T3, discarding T2 (lines 139-143). Excluding the middle tertile may reduce statistical power, and modeling tertiles as a categorical variable is generally less efficient than using TG/HDL‑C as a continuous variable. The authors should provide a clear justification for choosing tertiles over continuous modeling for the univariate regression analyses.
d) An FDR threshold of <0.0001 is noted (lines 152–154), but elsewhere (lines 262–263) an FDR <0.01 is mentioned. Please clarify these thresholds and their rationale. By the way, an FDR threshold of <0.0001 appears unusually stringent compared to commonly used thresholds (e.g., 0.05).
e) Mann–Whitney U tests were used for group comparisons (lines 191–192), but linear regression was also applied (lines 147–148). Did the authors check for normality? If data were normal, why mix parametric and non‑parametric methods without explanation?
Additionally, it would be helpful to clarify whether the blood samples used for lipid and metabolomic analyses were collected under fasting conditions.
5. Results
a) Table 1 and Supplementary Table 1 present baseline characteristics and include a very large number of variables. Many values in Supplementary Table 1 duplicate those in Table 1 without providing additional insight. While the data are comprehensive, several variables (for example, handgrip strength, myoglobin) are not directly relevant to the study’s aims. Including such extensive data distracts from the main focus of the paper. I recommend limiting these tables to variables directly relevant to MetS criteria and lipid metabolism, and streamlining the supplementary tables to include only variables that support the main results.
b) In Table 2, the columns labeled “5%” and “97.5%” appear to represent the lower and upper bounds of the 95% confidence interval. For clarity, I recommend explicitly labeling these columns as the 95% confidence interval.
c) Lines 198–206 and Table 3: the adjusted analyses show that TG/HDL‑C is not significantly different from TC/HDL‑C or non‑HDL/HDL, yet the manuscript continues to describe TG/HDL‑C as superior to all other ratios. While the results demonstrate that TG/HDL‑C has strong predictive performance and highlight metabolomic differences, they do not clearly establish its superiority over all other lipid ratios after adjustment, nor do they connect the metabolomic findings back to clinical utility or risk stratification. Please explicitly state in the Results that TG/HDL‑C is superior only to LDL/HDL after adjustment, not to all ratios.
In addition, were the pairwise tests between lipid ratios adjusted for multiple comparisons?
In Table 3, the unadjusted and adjusted values reported for TG/HDL‑C versus TC/HDL‑C and TG/HDL‑C versus non‑HDL/HDL appear to be exactly the same. Could you please check these values and confirm that they are correct? Furthermore, in Table 3, the two columns presenting AUC values for TG/HDL appear to repeat the same information three times. I recommend revising the table layout to eliminate repetition of information, and make it more concise.
d) Figure 2b and Table 5 both highlight similar metabolite classes (sphingomyelins, plasmalogens, phosphatidylethanolamines, monoacylglycerols, phosphatidylinositols, lysoplasmalogens) without explaining distinct contributions. Although presenting complementary analyses can be valuable, the overlap here is substantial, and the manuscript does not explain what unique insights each analysis provides. To avoid redundancy, please clarify why both analyses are included and what additional information each contributes. If Figure 2b is exploratory and Table 5 is confirmatory, this should be stated; otherwise, please streamline to avoid redundancy.
e) OPLS‑DA and univariate results are presented in Figure 2 and Table 4, respectively, and occupy a large portion of the Results section. However, these findings are not clearly linked back to clinical utility/prediction, which is central to the title and stated purpose of the study. It would be helpful to clarify whether the identified metabolites improve discrimination or risk stratification beyond TG/HDL‑C alone. If no such analysis was performed or intended, the title (and possibly the stated purpose) of the study may need to be revised to more accurately reflect the scope of the work.
f) Please ensure consistent terminology for clarity; the manuscript alternates between “TG/HDL” and “TG/HDL‑C.” Please use “TG/HDL‑C” throughout.
6. Discussion
a) The Discussion repeatedly portrays TG/HDL‑C as a robust and superior predictor despite adjusted ROC analyses showing no significant difference versus TC/HDL or non‑HDL/HDL (p = 0.166). This overstates the findings. Please temper the language to match the data.
b) Additionally, the Discussion interprets lipid class findings but does not critique the choice of tertiles versus continuous modeling.
c) The metabolomics discussion is lengthy and somewhat repetitive, and it does not directly address whether metabolomics improved clinical utility. Please clarify that these findings are exploratory mechanistic insights, not evidence of improved risk prediction.
7. Conclusions
The Conclusions section appears to overstate the strength of the evidence presented. For example, it describes TG/HDL‑C as a “simple, cost‑effective tool for early risk stratification” and suggests that the findings could serve as “therapeutic targets.” Given the cross‑sectional design and absence of prospective validation, these claims are not supported. No analyses were conducted to show that TG/HDL‑C or the identified metabolites improve patient outcomes or provide incremental predictive value beyond existing markers. In addition, the claim of novelty is not fully substantiated by the findings reported. The authors should also acknowledge that the metabolomics results are exploratory in nature and require confirmation in independent cohorts. Overall, I recommend toning down the Conclusions to more accurately reflect the scope and limitations of the study.
8. Other issues:
a) It appears that Reference 15 and Reference 18 refer to the same study. I also recommend reviewing the entire reference list for any inconsistencies in citation style and ensuring that the formatting is fully aligned with the journal’s guidelines.
b) The English and scientific writing are generally acceptable, but some sentences are overly long or complex, and there are minor grammatical errors and inconsistent abbreviations. A careful language edit would improve clarity and readability.
Author Response
Please see the attachement

Reviewer 3 Report
Comments and Suggestions for Authors
The article by Kano N. et al is aimed to elucidate available biomarker for earlier diagnostic and risk stratification of Metabolic Syndrome (MetS), overall metabolic disfunction associated with cardiovascular disorders, glucose intolerance and T2D development. The undoubted advantages of the work include the wide application of various modern mathematical approaches such as ROC/AUC-analysis, pairwise comparison of ROC-curves using DeLong’s test (Fig.1), multivariable analysis of high and low tertile groups in SIMCA resulted in orthogonal partial least squares-discriminant analysis (OPLS-DA) which allows to analyse metabolic profile associated with low and high tertiles with low and high TG/HDL ratio correspondingly (Fig.2 and Table 4) and finely linear regression analysis using selected metabolites as y- and tertiles (low vs high) as x-variables allowed to made functional enrichment analysis based on Wilcoxon sum of rank test and to avaluate 6 types of lipids changes of which are most significant between high and low tertiles (Table 5). Thus this article successfully combines modern mathematical statistics with good biochemical analysis of distinct alterations in lipid metabolism associated with TG/HDL ratio and suggests this index as biomarker of MetS for diagnosis and prognosis of this pathophysiological syndrome and also as a possible target for treatment. The impact of changes in lipid metabolism is discussed quite fully by the authors, who also note such limitations of their study as the limited demographic contingent, no information about diet, physical activity, medications used by participants of the study which could affect significantly the results obtained.
I would like also to add some comments. Considering ROC-curves in Fig.1 I should say that selection of TG/HDL index looks slightly artificial. According to corresponding ROC-curves all 4 indexes analyzed by the authors look like a good biomarkers of MetS and the differences are really negligible after adjusting for age, gender and BMI. Besides the curve non-HDL/HDL marked by light violet is absent. I should note that non-HDL/HDL is a very important index reflecting the ratio between the sum of atherogenic fractions of cholesterol to non-atherogenic and the risk of heart attack or stroke is very high if this index exceeds 4. The authors should correct this misunderstanding and make the curve visible otherwise they must exclude non-HDL/HDL index from parameters analyzed in the study.
The comments made do not detract from the good impression of the work, which is to be published in Metabolites, but should be regarded as a pioneering study and not as a final recommendation. Further research in this area is undoubtedly required to support recommendations for diagnosis and treatment of MetS.
Author Response
Please see the attachement

Reviewer 4 Report
Comments and Suggestions for Authors
The paper might be of interest. The reconsidered points were indicated.
- Is ‘metabolic syndrome’ in Title proper ‘metabolic syndrome without type 2 diabetes’?
- Diabetes and cardiovascular disease are not equivalent as the outcomes because diabetes is a risk factor for cardiovascular disease. Metabolic syndrome includes diabetes. The concept and rationale of study are difficult to understand.
- In adjustment, could the variables that have similar meanings to the outcome variable be used as the explanatory variables? For instance, can the BMI be used as an explanatory variable for MetS (including the factor of BMI) as the outcome variable.
- Written informed consent given to each participant could be stated in the Method section
- Treatment with the medications in the criteria could be concretely detailed.
- There is a gender difference in the characteristics of MetS. Please add the gender-stratified analysis.
- The NT-proBNP level was different between MetS and non-MetS participants. This could be fully discussed. The participants with cardiovascular disease were stated to be excluded.
- In Lines 273-274, there is the description that the finding of TG/HDL-C (not HDL) is consistent with the prior findings. If so, what is the novelty of this study? Please detail it.
- The study presented the AUC. How could the cut-off values be discussed?
- Could this finding of this study be applied to the prevention or clinical practice? Please discuss it more.
- TG, HDL-C, and TG/HDL-C could be respectively examined for the outcome.
- Similarly, LDL-C, non-HDL, LDL-C/HDL-C, and non-HDL-C/HDL-C could be respectively examined for the outcome.
- ‘LDL and HDL’ and ‘LDL-C and HDL-C’ were mixed throughout the text.
- The sampling conditions (fast or non-fast) could be stated.
- If TG was > 400 mg/dL, how were the LDL-C levels treated in the study?
- The assay methods of each lipid marker could be detailed.
- The assay performance could be also described.
- As well, the assay methods used for glucose, liver, kidney, and cardiac function could be disclosure.
- The terms of Abstract, Text, and table/Figure could be respectively abbreviated in the first use of terms. Tis is the general rule.
- In Abstract, what were the confounders in the expression of ‘adjusted’ AUC?
- In Table 1, each variable (i.e., lipid profiles) could have the unit.
- In Table 1, ‘HBA 1C’ could be expressed as ‘HbA1c’.
- In each Table and Figure, the abbreviations could be added in the footnote.
- In each Table and Figure, the decimal points could be unified.
- The abbreviations and full spellings were repeated overtime. For instance, MetS in line 39 and metabolic syndrome in Line 403. Please check such a phenomenon throughout the text.
- In Journal name of reference list, abbreviated and full spelled patterns were mixed.
- Please check the expression of page number in references.
Please check it again.
Round 2
Reviewer 1 Report
Comments and Suggestions for Authors
I want to congratulate the authors for their work and for the high quality of the revision. I wish them good fortune in their future work.
Author Response
We sincerely thank the reviewer for their encouraging feedback and kind words. We greatly appreciate the time and effort dedicated to reviewing our work and are grateful for the constructive comments that helped us strengthen the manuscript.
Reviewer 2 Report
Comments and Suggestions for Authors
The revised manuscript represents a clear improvement over the initial submission, with many of the substantive issues from the first review now addressed. The overall structure is more coherent, and the presentation of results is clearer. However, the analysis still relies primarily on comparing the lowest (T1) and highest (T3) tertiles of TG/HDL-C, while excluding the middle group (T2). Although this approach can be useful for highlighting differences in exploratory metabolomics, it is not the most appropriate method for evaluating overall associations or diagnostic performance. For methodological completeness, I recommend:
a) Performing an adjusted analysis treating TG/HDL-C as a continuous variable, including an assessment for potential non-linearity (e.g., using spline modeling).
b) Conducting a formal trend test across all three tertiles.
If these additional analyses yield results consistent with the current findings, please include a clear statement in the Limitations section indicating that the “T1 vs T3 comparison was employed primarily to maximize biological contrast in the exploratory metabolomics analyses, whereas the main conclusions are supported by and remain unchanged from the continuous-variable analysis."
Other issues:
a) The Introduction in the revised manuscript is overly long and could be made more concise without losing essential background. I recommend condensing the section by removing repeated explanations and streamlining the background, ensuring the last paragraph focuses only on the study’s unique rationale and objectives. This would improve readability and sharpen the manuscript’s focus.
b) Line 33: Rephrase “…revealing underlying perturbed metabolic pathways…” to “…revealing exploratory insights into metabolic pathways…”.
c) Lines 303-304: Soften “These findings support TG/HDL-C as a simple and accessible biomarker…” to “These findings suggest that TG/HDL-C may serve as a simple and accessible biomarker…”.
d) Ensure consistent variable names for all lipid ratios across both the text and tables.
Reviewer 4 Report
Comments and Suggestions for Authors
Some improvement was done. On the other hand, there was no treatment with drug medication for each patient. The were incomplete consideration of the secondary dyslipidemias for instance due to thyroid diseases. These would make the results reliable. Could the authors address such weak points of study?
Dear Editors; do you recommend rechecking the data of each participant in such addressed points? The results of drug-free population are useful.
